# Hypertensive Nephropathy: Unveiling the Possible Involvement of Hemichannels and Pannexons

**DOI:** 10.3390/ijms232415936

**Published:** 2022-12-14

**Authors:** Claudia M. Lucero, Juan Prieto-Villalobos, Lucas Marambio-Ruiz, Javiera Balmazabal, Tanhia F. Alvear, Matías Vega, Paola Barra, Mauricio A. Retamal, Juan A. Orellana, Gonzalo I. Gómez

**Affiliations:** 1Instituto de Ciencias Biomédicas, Facultad de Ciencias de la Salud, Universidad Autónoma de Chile, El Llano Subercaseaux #2801, Santiago 8910060, Chile; 2Departamento de Neurología, Escuela de Medicina y Centro Interdisciplinario de Neurociencias, Facultad de Medicina, Pontificia Universidad Católica de Chile, Santiago 8330024, Chile; 3Programa de Comunicación Celular en Cáncer, Facultad de Medicina Clínica Alemana, Universidad del Desarrollo, Santiago 7610658, Chile; 4Centro de Fisiología Celular e Integrativa, Facultad de Medicina Clínica Alemana, Universidad del Desarrollo, Santiago 7610658, Chile

**Keywords:** hypertensive nephropathy, connexin 43 hemichannel, gap junctions, pannexins 1 channels, oxidative stress, inflammation, Ca^2+^ dynamics

## Abstract

Hypertension is one of the most common risk factors for developing chronic cardiovascular diseases, including hypertensive nephropathy. Within the glomerulus, hypertension causes damage and activation of mesangial cells (MCs), eliciting the production of large amounts of vasoactive and proinflammatory agents. Accordingly, the activation of AT1 receptors by the vasoactive molecule angiotensin II (AngII) contributes to the pathogenesis of renal damage, which is mediated mostly by the dysfunction of intracellular Ca^2+^ ([Ca^2+^]_i_) signaling. Similarly, inflammation entails complex processes, where [Ca^2+^]_i_ also play crucial roles. Deregulation of this second messenger increases cell damage and promotes fibrosis, reduces renal blood flow, and impairs the glomerular filtration barrier. In vertebrates, [Ca^2+^]_i_ signaling depends, in part, on the activity of two families of large-pore channels: hemichannels and pannexons. Interestingly, the opening of these channels depends on [Ca^2+^]_i_ signaling. In this review, we propose that the opening of channels formed by connexins and/or pannexins mediated by AngII induces the ATP release to the extracellular media, with the subsequent activation of purinergic receptors. This process could elicit Ca^2+^ overload and constitute a feed-forward mechanism, leading to kidney damage.

## 1. Introduction

### 1.1. Chronic Kidney Disease, General Aspects

The kidney is commonly described as an excretory organ and is essential to maintaining systemic homeostasis [1,2]. Kidney-mediated blood filtering and modification of the ultrafiltration content by reabsorption and secretion, maintains pH, water, and electrolyte balance, contributing to regulating blood pressure and osmolality. The final product of the kidneys is urine, and its volume and composition can provide insights into body health status [2]. As mentioned, the kidneys’ central role is to eliminate waste materials ingested or produced by metabolism and to control the fluid body volume and electrolyte composition [3]. However, this organ can fail and when that happens, several deregulations in many organ functions can emerge. In this regard, chronic kidney disease (CKD) is a global health problem, the prevalence of which has increased [4]. Around 10% of adults suffer kidney damage in developed countries [5,6]. Currently, the estimated global prevalence of CKD is 697.5 million cases, and patients with end-stage renal disease (ESRD) who need renal replacement therapy range between 4.902 and 7.083 million [7,8].

CKD embodies different etiologies, including diabetic nephropathy, hypertensive nephrosclerosis, and glomerulonephritis. However, regardless of the causes, some morphological changes are common, including tubular necrosis and glomerular sclerosis [9,10,11] (Figure 1). The pathogenesis of CKD begins with a disturbance of the glomerular and tubulointerstitial compartments due to the release of cytokines from the glomerulus. This local inflammatory impairs tubular epithelial cell functions, causing excessive protein filtration at the injured glomeruli [11,12]. The above is then accompanied by tubulointerstitial ischemia at the glomerular lesion and the hyperfunction of the remaining tubules, promoting leukocyte, cytotoxicity, and fibrogenesis recruitment [11,12] (Figure 1). Altogether, these events could lead to acute renal failure, characterized by the abrupt and transitory decrease in renal function, inability to excrete waste products, and maintain the electrolyte balance, as well as generalized nephron dysfunction [13]. In contrast, patients with CKD usually develop progressive kidney damage typified by glomerular sclerosis or tubulointerstitial fibrosis, which eventually leads to end-stage renal disease (ESRD), and the last stage of this condition causes irreversible damage [14,15]. This phenomenon includes the progressive reduction in the glomerular filtration rate (GFR), due to increased nephron damage, which eventually triggers organ failure [6,16,17].

### 1.2. Role of Renin–Angiotensin System in CKD Development

Based on the renin localization, mainly in granular cells of the renal afferent arteriole, it is assumed that swelling or shrinking of the juxtaglomerular apparatus (JGA) is involved in the glomerular blood flow ascribed to the renin–angiotensin system (RAS). These cells make contact with the extraglomerular mesangium and macula densa, forming the JGA [18]. The unique juxtaposition of the macula densa to the afferent arterioles suggests the existence of mechanisms linking tubular function with the control of arteriolar contraction [18]. RAS is present not only in the juxtaglomerular apparatus and interstitium, but also in various segments of the nephron. Renal tubular RAS was recognized when the highest known angiotensin concentration in the body was measured in tubular fluid [18,19]. Subsequently, it was shown that all components of RAS are present in vascular smooth muscle cells of afferent arterioles and glomerular podocytes, suggesting the involvement of the intracellular renin–angiotensin system in local regulatory systems [18,20]. Hypertension is one of the most common risk factors for developing coronary artery disease, stroke, heart failure, peripheral arterial disease, vision loss, dementia, and CKD. The prevalence of hypertension among adults will escalate to more than 1.56 billion by 2025 [12,21]. Hypertensive nephropathy linked to hypertension is the second cause of terminal CKD, and, nowadays, the cost of its treatment exceeds the budget of health systems in many countries [18,19]. The inflammatory injury and tissue remodeling evoked by CDK have been attributed to angiotensin II (Ang II), the principal peptide in RAS [22].

AngII regulates blood pressure, intravascular volume, and electrolyte balance by acting on the kidneys and adrenal glands [22]. Treatment with high concentrations of AngII induces the expression of chemokines, adhesion molecules, and pro-inflammatory cytokines (interleukin- 1β [IL-1β] and tumor necrosis factor-α [TNF-α]) [15,23,24]. The latter is accompanied by infiltration of macrophages (positive ED-1) [25], and tubular overexpression of osteopontin, a macrophage chemoattractant and adhesion molecule [25]. All these events are intimately associated with AngII-induced kidney damage linked to redox imbalance [26,27]. AngII increases the expression of nicotinamide adenine dinucleotide phosphate (NADPH) oxidase (NOX), one of the essential enzymes in the generation of reactive oxygen species (ROS) [28,29]. Indeed, the impairment of redox balance plays a decisive role in several chronic diseases, including the progression of CKD [6,30] (Figure 2). Given that intrarenal RAS is critical for the pathophysiology of hypertensive nephropathy [6,31], the administration of AngII has been used for modeling kidney damage. In addition to providing a clinically relevant model for systemic hypertension, exposure to AngII leads to progressive and chronic nephron damage that recapitulates events occurring in ESRD [27,31,32]. Due to the diverse effects of RAS, RAS inhibitors (including blockers of angiotensin converting enzyme, angiotensin receptors, renin/prorenin receptors, and renin release) reduce blood pressure and increase renal blood flow and GFR, decreasing the progression of renal tissue damage [17].

### 1.3. Role of Mesangial Cells in CKD

The renal corpuscle comprises the Bowman’s capsule and the glomerulus, both being fundamental structures for blood filtration. The Bowman’s capsule plays a vital role in the structural and functional stability of the glomerulus, allowing it to fulfill its filtering function successfully [20,33]. Structurally speaking, the mesangium consists of a network of mesangial cells (MCs) that constitute between 30–40% of the glomerular cell populations. They control the capillary dilation, the regulation of the GFR, and the synthesis and degradation of extracellular matrix proteins [33]. Moreover, they have contractile and phagocytic properties [34]. Interestingly, cells from the renin lineage repopulate the niche of damaged glomerular MCs that have undergone mesangiolysis via the experimental model of mesangial proliferative glomerulonephritis. Still, the mechanism and molecules involved in the above phenomenon remain to be defined [35]. On the other hand, MCs stimulated with AngII produce ROS [14,36,37], synthesize and release IL-1β, TNF-α, and chemokines, such as the macrophage chemoattractant protein (MCP-1) and the transforming growth factor β1 (TGF-β1). These inflammatory mediators favor fibrosis, reducing renal blood flow, permeability, and eventually glomerular filtration [14,20,27,33,37,38,39]. Transcription factors NF-κB and AP-1 command these alterations and increase the synthesis and release of extracellular matrix proteins, such as type IV collagen, laminin, and fibronectin (Figure 2). The latter result in the formation of mesangial nodules and lesions at the interstitium, hindering the adequate function of the glomerulus [27,33,37,38,39]. As MCs undergo progressive damage at the glomerulus, they may serve as a potential target for understanding the pathogenesis of chronic nephron lesions and subsequent CKD [14,33,36,38]. In this review, we discuss how MCs could contribute to glomerular dysfunction following hypertensive nephropathy, focusing on the signaling pathways mediated by two families of plasma membrane channels: hemichannels and pannexons.

## 2. Connexins and Pannexins

Gap junctions (GJs) are aggregates of ten to thousands of individual intercellular channels that permit the cytoplasmic exchange between adjacent cells. They allow the diffusion of ions, small metabolites (e.g., nicotinamide adenine dinucleotide: NAD+, glucose, lactate, and glutamate), and second messengers (e.g., cyclic adenosine monophosphate [cAMP] and inositol trisphosphate [IP3]) between adjacent cells [40,41,42]. These connections support several biological processes in the animal kingdom, such as the propagation of electrical and chemical signals, which are crucial for cell coordination and survival. GJ channels are formed by integral membrane proteins called connexins, whose family has 21 isoforms in humans that are named according to their molecular weight [42,43]. All connexins feature four transmembrane domains, two extracellular loops, each with three highly conserved cysteine residues, an intracellular loop, and the C- and N-terminal ends on the cytoplasmic domain [42]. Six connexin monomers assemble around a central pore to form a hemichannel (HC) or connexons, whereas the apposition of two connexons from adjacent cells creates an intercellular GJ channel [42]. Cells expressing more than one type of connexin could eventually form heteromeric connexons. Likewise, cells expressing different connexons could form heterotypic GJs [42].

HCs act as free channels in the plasma membrane, favoring the exchange of small molecules and ions between the cytosol and the extracellular space [43,44] (Figure 3). Al-though initially they were catalogued as non-selective channels, nowadays, the evidence seems to indicate the opposite [45,46,47,48]. For example, Cx32 HCs show a slight preference for anions, while Cx40 and Cx43 have a high selectivity for cations [49,50,51]. In the same way, selectivity for ATP is 300-fold higher in Cx43 than in Cx32 HCs; for ADP, glutamate, and glutathione, the selectivity is 20-fold higher in Cx43 than in Cx32 HCs, while adenosine is 10-fold more permeable in Cx32 than Cx43 HCs [42,52]. The physiological opening of these channels is critical for the proper function of multiple biological processes, such as neurite outgrowth [53], neutrophil migration [54], and regulation of acid–base homeostasis in the kidney [55], among others. The HC opening is tightly regulated and relies on changes in voltage, intracellular/extracellular Ca^2+^ concentration ([Ca^2+^]_i_) [56], pH [57], post-translational modifications (e.g., phosphorylation [58] and S-nitrosylation [59]), and protein interactions [42]. Nevertheless, persistent activation of HCs has been associated with the genesis and progression of diverse pathological stages [60,61,62]. At least three mechanisms have linked the exacerbated opening of HCs with cell dysfunction and damage [63]. Firstly, the uncontrolled entry of Na^+^ and Cl^−^ through HCs may result in osmotic and ionic imbalances coupled to further cell swelling and plasma membrane breakdown [63,64,65]. Second, some HCs are permeable to Ca^2+^ [66,67], which could permit its influx to the cytosol during pathological conditions. The direct or indirect increase in [Ca^2+^]_i_ mediated by HCs could lead to Ca^2+^ overload and consequent induction of different proteases, phospholipases, and other hydrolytic enzymes, as well as oxidative stress and caspase activation [68,69]. Finally, exacerbated HC opening may trigger the release of high amounts of potentially toxic molecules to neighboring cells, such as glutamate, in the case of the brain [63,70].

Pannexins (Panxs) belong to a family of transmembrane proteins that bear structural homology with innexins, the GJ protein family of invertebrates [71]. Although connexins and pannexins do not share significant sequence homology, both families exhibit similar transmembrane topology: four transmembrane domains with two extracellular loops and intracellular N- and C-terminals. Unlike connexins, which have several extracellular cysteine residues, pannexins have only two cysteines and a highly glycosylated aspartate in the second extracellular loop, which would hinder GJs formation (see below) [72,73] (Figure 3). Three pannexin isoforms have been identified to date: pannexin-1 (Panx1) (~48 kDa), is the most ubiquitous member of its family, broadly expressed in the brain, muscle, and other tissues [74]; pannexin-2 (Panx2) (~45 kDa), predominantly expressed in the brain; and pannexin-3 (Panx3) (~73 kDa), found in skin, bone, cartilage, skeletal muscle, and blood vessels [72,73,75,76]. Pannexin monomers oligomerize around a central pore in heptamers (Panx1, Panx3) or octamers (Panx2) called pannexons, allowing the exchange of ions and small molecules between the cytoplasm and the extracellular milieu [71,72,77]. Panx1 channels have been historically described as non-junctional membrane channels; nevertheless, recent evidence suggests that Panx1 can form GJs channels due to a differential glycosylated state at its extracellular loop [78]. Sequence analysis of Panx1 reveals that the Asp254 residue can be gradually glycosylated, which yields three different Panx1 bands in Western blot analysis: Gly-0 (non-glycosylated), Gly-1 (several mannose glycosylation), and Gly-2 (complex sugar glycosylation) [78]. Mutagenesis studies show that these glycosylations facilitate Panx1 trafficking but are not essential for the arrival of Panx1 to the plasma membrane or the function of a GJ channel. However, the Gly-2 variant of Panx1 is predominant on the surface [72,78].

Panx1 channels are associated with the regulation of vascular tone and flow [79], mucociliary clearance in the airway epithelial [80], phagocytosis of apoptotic cells [81], activation of the inflammasome, and recruitment of immune cells [82,83]. Similar to HCs, the persistent opening of Panx1 channels has been proposed to be harmful to cells, contributing to cell death in episodes of ischemia [84] or facilitating viral infection [85]. Panx1 channels have been extensively studied as crucial pathways for releasing ATP and pivotal players for purinergic receptor-mediated intracellular Ca^2+^ wave propagation [86,87,88]. Indeed, Panx1 channels release ATP at resting membrane potential and under physiological extracellular concentrations of divalent cations (Ca^2+^ and Mg^2+^) [73]. Furthermore, multiple direct mechanisms activate these channels, including mechanical stimulation [89], caspases-dependent cleavage of the C-tail of Panx1 [81], hypotonicity, ischemia [73], or a secondary response to the activation of receptors ionotropic as P2X4, P2X7, and NMDAR, which are permeable to cations, including Na^+^, K^+^, and Ca^2+^ [83,88,90], or metabotropic receptors such as P2Y1, P2Y2, and P2Y6 [90], protease-activated receptor-1 (PAR-1), bradykinin B_2_ receptor (BDKRB2) [91], histamine H_1_ receptor (HRH1), and α_1D_-adrenergic receptors (α_1D_-AR) [92].

## 3. Connexins and Pannexin-Based Channels in the Kidney

The kidney expresses nine connexin isoforms, including Cx26, Cx30, Cx30.3, Cx32, Cx37, Cx40, Cx43, Cx45, and Cx46 [93,94,95,96] (Figure 4). The evidence indicates that these connexins could play a pivotal role in important aspects of renal function, such as the maintenance of acid–base homeostasis, hydroelectrolyte balance, regulation of blood pressure, and functional processes of the nephron: glomerular filtration, reabsorption and secretion of metabolites, water reabsorption, and renin secretion [93,94,95,96].

### 3.1. Glomerular Connexin and Pannexin-Based Channels

Cx43 is found in the endothelium of the pre-glomerular and post-glomerular regions. In that place, Cx43 builds GJs between endothelial cells, SMCs, and among endothelial cells and SMCs. Moreover, the myoendothelial GJ coupling occurs more frequently in the juxtaglomerular segment of afferent arterioles than in the proximal region and interlobular arteries [94]. The secretory renin-producing cells mainly express Cx40, Cx37, and Cx45 [55,93,97,98]. The role of Cx40 in renin secretion and blood pressure regulation has been established [99,100]. Cx40-deficient mice exhibit hyperreninemia and hypertension because of overactivation of the renin–angiotensin–aldosterone system [101,102,103]. The evidence suggests a decreased coupling between renin-producing cells and neighboring cells. The latter could impair the increase in [Ca^2+^]_i_ to renin-secreting cells [102,103,104]. Within the glomeruli, the intraglomerular MCs express mainly Cx40 [105], Cx45 [98], Cx43 [106], and Cx37 in the vascular pole [98], while Cx43 is expressed by podocytes [106]. Ultrastructural studies have shown the presence of GJs between different cell populations at the glomerulus [107] with strong levels of Cx40 in the entire intraglomerular mesangium. Cx37 shows expression only in MCs from the vascular pole of the glomerulus [97,108] and GJs composed of Cx43 have been detected between the podocytes [109] (Figure 4).

Although Panx1 represents the most ubiquitous isoform in mammal tissues, it is poorly detected in the kidney tissues [110]. Panx1 expression has been shown in renal vasculature, juxtaglomerular cells [111,112,113], and α-actin-positive SMCs in the afferent arteriole [113]. No data on Panx2 expression have been described in the kidney, whereas Panx3 has been observed in juxtaglomerular cells [114]. Panx1 channel opening reduces renin secretion from renin-expressing cells by increasing [Ca^2+^]_i_, which further inactivates Ca^2+^-sensitive adenylate cyclase. However, the mechanisms associated with cAMP and the subsequent inhibition of renin secretion are poorly understood [111]. Furthermore, P1 and P2X _(1-2-4-7)_ and P2Y _(1-2)_ receptors show expression in the renal vasculature [115], where the P2X_1_ receptor increases vascular resistance in the afferent arteriole [115] and promotes renin secretion from juxtaglomerular cells [116]. Conversely, P1 receptors mediate tubuloglomerular feedback and suppresses Ca^2+^-dependent renin secretion [117]. Furthermore, it has been seen that extracellular ATP activates P2X_7_Rs, which are permeable to cations, including Na^+^, K^+^, and Ca^2+^ [88]. In turn, the activation of P2X_7_R leads to Panx1 channel opening [15] (Figure 4).

### 3.2. Tubular Connexin and Pannexin-Based Channels

Under physiological conditions, epithelial cells of the proximal tubule express Cx26 and Cx32, where they colocalize with megalin throughout the proximal tubule segment [98,118]. Additionally, in this renal structure, Cx37 [119] and Cx43 [118] have also been detected in the above cells. Interestingly, in diabetic animals, both Cx26 and Cx43 increase their expression in epithelial cells of the proximal tubule [120]. On the other hand, in the thin ascending limb of the Henle loop and intercalated cells of the collecting duct, the presence of functional HCs formed by Cx30.3 has been confirmed [55]. Purinergic receptors are expressed in this segment and their activation could rely on ATP released through Cx30.3 HCs [55]. On the other hand, Cx37 has been observed in the thick ascending limb of the Henle loop, with its expression being regulated by salt intake, indicating that it likely contributes to Na^+^ reabsorption [119]. Furthermore, the expression of Cx30 in distal and collecting tubules has also been observed [121]. The opening of Cx30 HCs allows the release of ATP, which reduces the reabsorption of NaCl, thereby increasing natriuresis. Relevant to the above, animals deficient in Cx30 excrete low levels of Na^+^, causing an increase in blood pressure due to salt retention [122] (Figure 4).

Immunodetection studies have shown the presence of Panx1 in different kidney structures, such as polarized epithelial cells, specifically in apical membranes of proximal tubules, thick descending limbs, and collecting ducts [111,112,113]. This strategic localization suggests that Panx1 regulates the release of ATP from epithelial cells towards the tubular lumen, contributing to renal hemodynamics and tubular salt and water transport [113,123]. Supporting this idea, high levels of extracellular ATP have been found in proximal tubule cells, which correlates with the increased expression of Panx1 and P2Y_1_ receptors [124,125]. Interestingly, the activation of P2Y_1_ receptors in the proximal tubules inhibits bicarbonate reabsorption [124,125]. Here, Panx1 could contribute to the release of ATP, similar to the way Cx30.3 HCs contribute in the proximal tubules [55,126]. Panx1 is also expressed in the principal and intercalated cells in the collecting tubule, which regulates water reabsorption and acid–base balance, respectively [127]. ATP released by Panx1 channels in principal cells inhibits the epithelial Na^+^ channel-mediated Na^+^ reabsorption linked to Ca^2+^ signaling [127,128,129] (Figure 4).

## 4. Cx43 and Panx 1 in Hypertensive Nephropathy

In pathological conditions such as hypertension, levels of renal connexins change. For example, in the two kidneys-one clip model (2K1C), an increase in Cx43 mRNA and protein levels in the glomerulus was observed [130]. A recent study by Abed and colleagues found that Cx43 augments its expression at the early stages of hypertension induced by obstructive nephropathy in the renal cortex of diseased mice [131]. Likewise, ablation of Cx43 improves the renal function of CKD induced by hypertension. The latter suggests that Cx43 contributes to renal cortex damage, and, therefore, its inhibition could be considered a therapeutic target in CDK models [131]. Also, Cx43 is selectively regulated in the vascular small muscle cells of renin-dependent models of hypertension, the AngII activation of the ERK and NF-κB pathways being crucial in this phenomenon [132]. On the other hand, after silencing Cx43 expression, aldosterone-induced apoptosis in cultured podocytes was partially attenuated, suggesting that Cx43 was involved in podocyte apoptosis, induced by aldosterone [131,133]. Toubas and collaborators observed that three different CKD models boosted the immunodetection of Cx43 in glomerulus and tubules, postulating that inflammation could be pivotal in the rise in Cx43 levels in the damaged kidney [134]. Similarly, using a model of renal fibrosis developed in HK2 cells, Potter and coworkers demonstrate that collagen I (an early marker of kidney disease) and TGF-β1 elevate the release of ATP via Cx43 HCs [135]. Furthermore, Yu and coworkers employed another HK2 cell model, and observed that silencing Cx43 expression prevented cisplatin-induced ferroptosis, as well as cell apoptosis [136] (Table 1).

TNF-α and IL-1β reduce intercellular communication mediated by GJs and increase the activity of Cx43 HCs in astrocytes [137]. Interestingly, cultures of proximal tubule cells treated with metabolic inhibitors or pro-inflammatory cytokines display an increased activity of HCs [118,138]. Moreover, in vivo administration of AngII for three weeks or less causes a transitory increase in inflammation, ROS, and Cx43 in the renal cortex, which is associated with renal damage. However, these alterations become irreversible and irreparable if AngII is administered for extended periods (four or more weeks), amplifying inflammatory and oxidative responses along with kidney cell damage [139]. These data support Cx43 as a potential new mediator of renal disease involved in central processes of inflammation and fibrosis, while its inhibition even after the initiation of the disease attenuates renal damage and preserves renal function in animal models of vascular, tubular, and glomerular CKD [140]. The inhibition of Cx43 likely represents a promising future therapeutic option against nephropathy. In summary, the available data highlight the promising beneficial effect of Cx43 inhibition on inflammation, tissue integrity, and fibrosis [141,142] (Table 1).

**Table 1 ijms-23-15936-t001:** Connexin 43 in hypertensive nephropathy.

Renal Hypertension Model	Experimental Model	Effect on Cx43	Renal Site	Technique	Reference
2K1C	KI32 mice	Without changes	G, aa, ila	IF, WB	Haefliger et al., 2006 [130]
Hypertension-induced CKD	RenTg mice	↓	Rc	qPCR, IF	Abed et al., 2014 [131]
Uninephrectomized, 1%NaCl and aldosterone infusion	SD rats	↑	Po	IF, WB, DCFDA fluorescence	Yang et al., 2014 [133]
RenTg, anti-glomerular basement membrane, unilateral ureteral obstruction	Mice	↑	G, T	rt-PCR, IF	Toubas et al., 2011 [134]
Collagen I and TGFβ1 treatment	HK2 cells	↑	PT (cells)	Carboxyfluorescein dye uptake	Potter et al., 2021 [135]
The cisplatin-induced kidney injury model	HK2 cells, mice	↑	Rc	IF, WB, IHC	Yu et al., 2021 [136]
ATP depletion	hPTC	↑	PT (cells)	IF, WB, fluorescein dextran dye uptake	Vergara et al., 2003 [118]
High glucose	hCDC	↑	CD	IF, WB, Lucifer yellow dye uptake	Hills et al., 2006 [138]
Angiotensin II infusion	SD rats	↑	Rc	WB	Gómez et al., 2019 [139]
TGFβ1 treatment	hPTECs, HK2 cells	↑	PT (cells)	Carboxyfluorescein dye uptake	Price et al., 2020 [142]

Different study models of hypertensive nephropathy and its effects on connexin 43. Abbreviations: 2K1C, two kidney-one clip model; aa, afferent arteriole; CD, collecting duct; G, glomerulus; hCDC, human collecting duct cells; HK2, clonal tubular epithelial cells; hPTC, human proximal tubule cells; ila, interlobular arteriole; Po, podocytes; PT, proximal tubules; Rc, renal cortex; SD, Sprague–Dawley; T, tubules. ↑ indicates that there is greater expression or protein levels; ↓ less expression or protein levels.

The AngII-mediated activation of AT1Rs is critical for the progressive deterioration of glomerular function, which contributes to the inflammatory and oxidative damage observed in renal diseases [143,144]. In this context, the impairment of Ca^2+^ buffering leads to podocyte cytoskeletal disorganization, foot process effacement, disruption of the slit diaphragm, and proteinuria, which are critical for the development of diabetic kidney diseases [145]. The AngII half-life lasts about a few hours [15,146]; however, soon after AngII binds to the AT1R [Ca^2+^]_i_ increases [147,148]. In this sense, those responses might be enough to increase the open probability of Panx1 channels [15,149], and the further release of ATP to the extracellular milieu [15,150]. As previously demonstrated in the nervous system, the ATP released as a result of the [Ca^2+^]_i_–induced Panx1 channel opening likely contributes to the progression of cell death. Critical in perpetuating this phenomenon could be the P2X receptor activation, as it leads to persistent Ca^2+^ entry and subsequent stimulation of intracellular toxic cascades [151].

In the last decade, various studies have linked the uncontrolled opening of Panx1 channels with some diseases [72]. Moreover, although not yet demonstrated, it is possible that Panx1 channels could be permeable to Ca^2+^, which possibly result in [Ca^2+^]_i_ overload and the consequent production of free radicals, lipid peroxidation, and plasma membrane damage [152]. Alternatively, uncontrolled Panx1 channel activity could also trigger the release of molecules that, at high concentrations, may be toxic to surrounding cells, such as ATP [152]. In other cell types, the increase in [Ca^2+^]_i_ raised the activity of Panx1 channels allowing the release of ATP and further activation of P2X_7_ and P2Y_1_ receptors [90], where cytokines could also perpetuate Panx1 channel opening [153,154].

On the other hand, more recent evidence has proposed that Panx1 channels increase [Ca^2+^]_i_ upon stimulation with TNF-α [155], which could lead to the production of free radicals, lipid peroxidation, and plasma membrane damage [152]. Panx1 has been detected in different areas of the kidney [113,156], and palmitic acid stimulates ATP release through the Panx1 channels in human renal tubule epithelial cells (HK-2) by a mechanism involving the activation of caspase-3/7 induced by mitochondrial ROS [155]. Indeed, another study shows that danger signals from necrotic tubular epithelial cells could activate the NOD-like receptor protein 3 (NLRP3) inflammasome in macrophages through the TLR2/caspase 5/Panx1 axis during acute kidney injury (AKI). Relevantly, Panx1 cleavage evoked by caspase 11 was crucial for facilitating the ATP release and posterior activation of the NLRP3 inflammasome during ischemia/reperfusion-induced by AKI [157,158]. Recent work of our group revealed that in AngII-stimulated mesangial MES-13 cells, a similar feed-forward mechanism occurs, similarly between Panx1 channels, Cx43 HCs, and/or P2X_7_Rs [15]. Accordingly, the blockade of the above protein channels dramatically prevented the AngII-induced increase in oxidative stress and release of cytokines [15].

It is known that high extracellular ATP levels contribute to the transformation of MCs and promote renal injury in AngII-dependent hypertension [159,160] (Figure 5). The latter may induce a self-perpetuating mechanism characterized by the influx of extracellular Ca^2+^ and IP_3_ receptor-dependent release of Ca^2+^ stored in the endoplasmic reticulum. This could occur in the companion of the Panx1 channels-dependent release of cytokines, all these processes being simultaneous, as previously demonstrated in other cell types [161,162] (Figure 5). However, it remains unknown whether Panx1 is present in native MCs and if Panx-1 channels and/or Cx-43 HCs mediate the release of ATP associated with glomerular inflammation or fibrosis, as seen in the example for podocytes and other cells [163,164].

## 5. Conclusions and Perspectives

For years, hypertensive kidney disease has been considered a disorder limited to afferent arterioles and glomeruli in which the mechanical stresses induced by hypertension, RAS stimulation, and activation of resident fibroblasts were thought to be crucial in underlying kidney damage [21]. Findings discussed in this review support activation of MCs-Cx43 HCs and Panx1 channels as potential contributors to glomerular abnormalities observed following hypertensive nephropathy. The opening of these channels could be the hidden link between mesangial dysfunction and impaired kidney function. Recent development of specific Cx43 HCs and Panx1 channel-blocking peptides will enable future study of molecular mechanisms underlying Cx43 and Panx1 signaling in renal functions.

Despite existing tools, such as connexin KO mice and connexin-blocking peptides, knowledge about connexin implications in renal physiology is still limited [165]. Moreover, whether required communication for renal homeostasis occurs via GJs or HCs is currently poorly understood, and the role of the pannexin family in renal physiology is still unclear. However, recent studies reported their involvement in several diseases in humans and rodents [165]. The use of Panx1-deficient mice would be of significant interest to increase knowledge of Panx1 channel biology in the kidney [165]. Future studies will clarify whether MCs connexin HCs and pannexon opening evoked by AngII occurs with glomerular damage. Understanding how HCs and pannexons participate in the impairment of glomerular cell cross-talk during hypertensive nephropathy is potentially critical for developing pharmacological therapies.

## Figures and Tables

**Figure 1 ijms-23-15936-f001:**
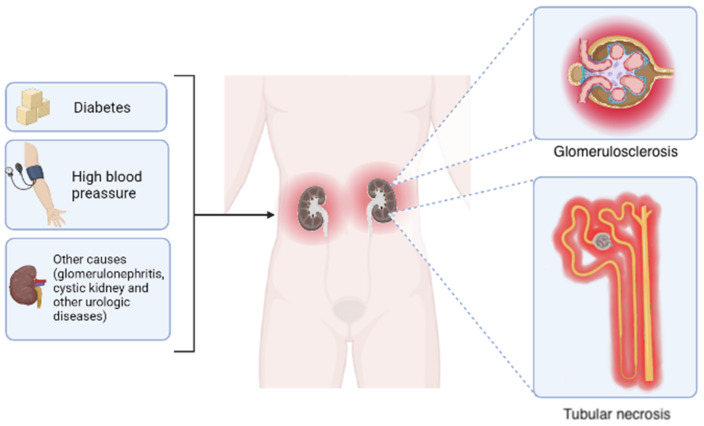
Primary causes and morphological outcomes of chronic kidney disease. Although the leading causes of chronic kidney disease include diabetes and hypertension, other disorders such as glomerulonephritis, cystic kidney disease, and diverse urologic diseases also contribute in a minor proportion. Regardless of the etiology, the progressive reduction in glomerular filtration rate occurs accompanied by two common histological changes: glomerular sclerosis and tubular necrosis. Elaborated in BioRender.com (accessed on 15 October 2022).

**Figure 2 ijms-23-15936-f002:**
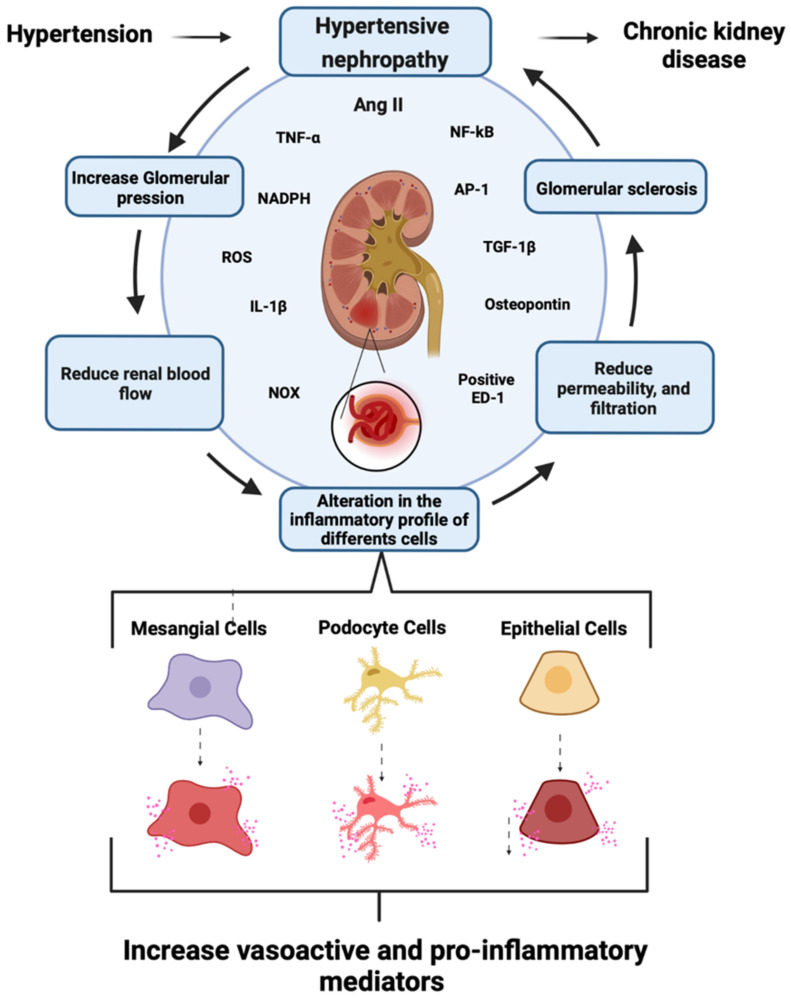
Schematics showing general aspects of the generation of chronic kidney disease mediated by hypertensive nephropathy. At the renal level, arterial hypertension increases intra-glomerular pressure, leading to reductions in renal blood flow and alterations in the inflammatory profile of mesangial cells, podocytes, and epithelial cells. The epithelial cells boost the production of vasoactive and pro-inflammatory mediators, triggering a reduction in glomerular permeability and filtration. Consequently, hypertensive nephropathy can contribute to CKD. Renal impairment would then be associated mainly with the persistent production of AngII, which could augment the expression of proinflammatory cytokines [IL-1β and TNF-α] and the NOX-mediated generation of ROS. These processes likely result in macrophage infiltration and tubular overexpression of osteopontin, favoring the production of TGF- β1 and activation of NF-κB and AP-1. Elaborated in BioRender.com (accessed on 15 October 2022).

**Figure 3 ijms-23-15936-f003:**
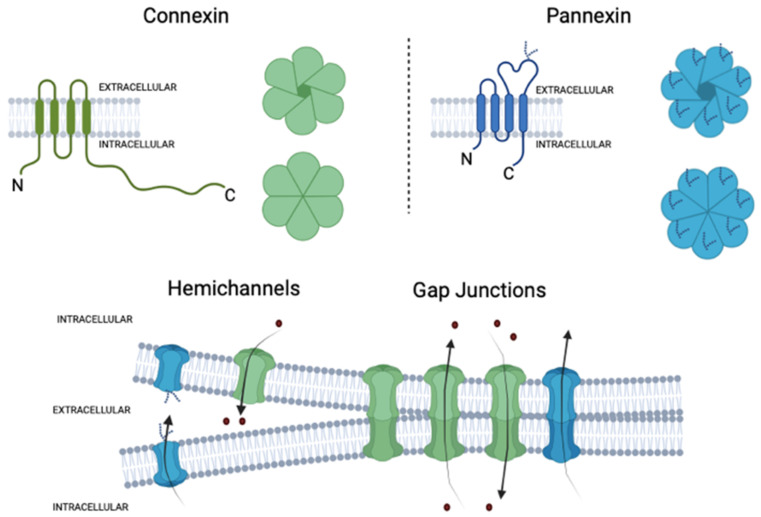
Structural organization of connexin and pannexin-based channels. Connexins and pannexins have four transmembrane domains (TM), two extracellular loops (EL), one cytoplasmatic loop (CL), and both the N- and C-terminus are localized on the intracellular side. The glycosylated extracellular loop of pannexins is also shown. Hemichannels or connexons are formed by the oligomerization of six subunits of connexins around a central pore, while pannexons are composed of seven pannexin subunits in the case of Panx1. Under normal conditions, the opening of hemichannels and pannexons is tightly regulated to fulfill several biological processes. Hemichannels and likely pannexons can also dock each other in junctional membrane interfaces to form cell-to-cell intercellular channels, termed gap junction channels. Elaborated in BioRender.com (accessed on 15 October 2022).

**Figure 4 ijms-23-15936-f004:**
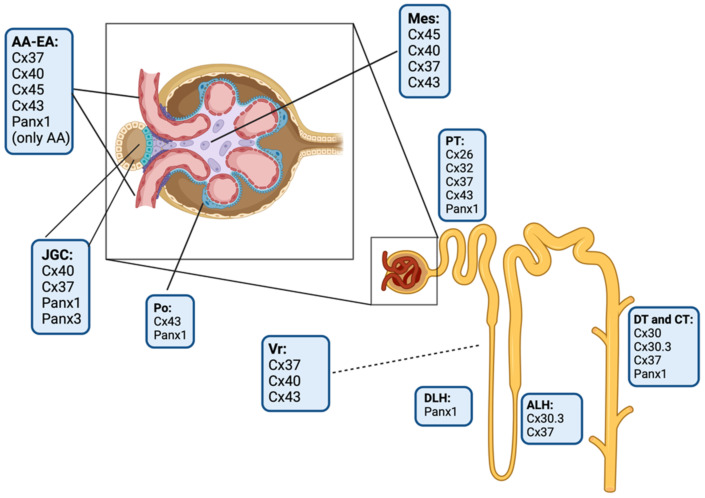
Schematic drawing of the localization of connexin and pannexin isoforms in the kidney. AA, afferent arteriole; EA, efferent arteriole; JGC, juxtaglomerular cells; Po, podocytes; Mes, mesangial cells; PT, proximal tubules; DLH, descending limb of loop of Henle; ALH, ascending limb of loop of Henle; DT, distal tubules; CT, collecting tubules; and Vr, Vasa recta. Elaborated in BioRender.com (accessed on 15 October 2022).

**Figure 5 ijms-23-15936-f005:**
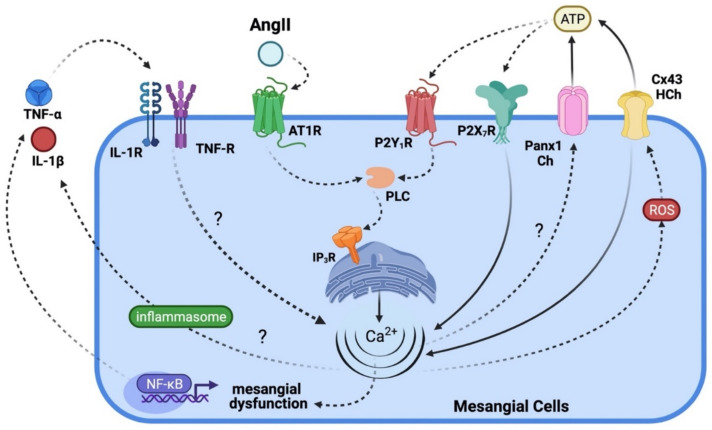
Scheme of possible signaling pathways involved in regulating of Cx43 hemichannels and Panx1 channels in mesangial cells stimulated with AngII. High AngII concentrations (10^−7^ M) activate angiotensin type 1 receptors (AT1R) in MCs, causing the PLC-mediated stimulation of IP3 receptors and further release of Ca^2+^ stored in the endoplasmic reticulum, NF-κB-dependent release of proinflammatory cytokines such as TNF-α and IL-1β, as well as the formation of reactive oxygen species (ROS). The resulting increase in [Ca^2+^]_i_ can also stimulate Panx1 channels and Cx43 hemichannels, eliciting the subsequent release of ATP to the extracellular compartment. Then, ATP activates P2X_7_ and P2Y_1_ receptors that, together with Cx43 hemichannels, permit a drastic increase in Ca^2+^ influx. As already seen in other systems, the increase in [Ca^2+^]_i_ promotes the expression and release of pro-inflammatory cytokines such as TNF-α, IL-1β via the activation of the inflammasome, and the generation of ROS. The release of ATP and influx of Ca^2+^ establish a positive feedback loop. Of note, alterations in [Ca^2+^]_i_ homeostasis mediated by Cx43 hemichannels or Panx1 channels could affect diverse aspects of MCs function (e.g., morphology and pro-inflammatory profile). Solid lines depict fluxes of molecules through channels, whereas dashed lines indicate activation or induction. Elaborated in BioRender.com (accessed on 15 October 2022).

## Data Availability

Not applicable.

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
