# Peer review of "Hypertensive Nephropathy: Unveiling the Possible Involvement of Hemichannels and Pannexons"

_ijms, 2022, doi:10.3390/ijms232415936_

Round 1
Reviewer 1 Report
In this review, the authors describe the expression of Cxs and Panx1-based hemichannels (HCs) (also known as connexons and pannexons) by the kidney and revise the involvement of these HCs in renal damage during hypertensive nephropathy. They elaborate a hypothesis focusing on the deleterious effects of HCs expressed by mesangial cells during renal hypertension where the roles of angiotensin II, ATP and calcium ions are highlighted. The review is timely and interesting and raises a topic that is increasingly addressed by the scientific community. Likewise, although described injurious processes are framed in the context of hypertensive nephropathy, they could represent a common mechanism underlying several nephropathies. Anyway, although the manuscript contains valuable information, there are some conceptual aspects described in the text (especially in the introduction) that should be improved to generate an acceptable manuscript since they reflect whether the authors understand key aspects of renal physiology.
Abstract
The abstract contains many details (which are subsequently developed in the text) that distract from main messages; in fact, it would be preferable to summarize the background and to include the aim of the review at the end of the abstract.
1. Introduction
The content described in lines 42-70 should be re-written or almost deleted. This is the main weakness of the manuscript. The kidney is a vital organ that regulates homeostasis of the internal media. Be aware that, “ultrafiltration, reabsorption, secretion and excretion” represent the functional processes developed by the nephron (the functional unit of the kidney) to maintain the homeostasis of the internal media. In line with this, the kidney eliminates those products that are required to be removed from the body and conserves the ones necessary to keep in order to successfully accomplishing the maintenance of internal medium homeostasis (volume, osmolality, arterial pressure, etc). Functional processes occurring in the nephron (ultrafiltration, reabsorption, secretion and excretion) and homeostatic functions of the kidney (regulation of the internal milieu including pH, osmolality, volume, arterial pressure) are not rigorously differentiated over the text. In addition, the description of renal structures should follow a logical sequence; i.e., lines 48-49: “Each nephron consists of a glomerulus, proximal and distal convoluted tubule, and loop of Henle.” Why to mention loop of Henle after distal convoluted tubule? Is the information about cortical and juxtaglomerular nephrons necessary for the revision (lines 51-53)? Similar question about the description about the renal epithelium (lines 62-68). These descriptions focus on some structural aspects, not related to the function they achieve and are incompletely assessed, indeed, it would be very difficult to resume and hierarchize the kidney structure and function in few paragraphs.
One possibility is to initiate the review by line 69, describing main homeostatic functions of the kidney (“regulation of the homeostasis of the internal media”, i.e., regulation of the volume, hidro-electric composition, osmolality, pH, arterial pressure) and then introducing to CKD, hypertension, role of ANGII and finally describe mesangial cells. Accordingly, the content of lines 54-61 should be relocated when referring to MCs.
Consider revising and citing: 10.1161/CIRCRESAHA.121.318064
2. Connexins and Pannexins
3. Connexins and Pannexins-based channels in the kidney
Glomerular connexin and pannexin-based channels
Lines 245-248: functional processes of the nephron (reabsorption and secretion of metabolites, glomerular filtration, water reabsorption, and renin secretion) should be differentiated from general homeostatic functions of the kidney (acid-base homeostasis; regulation of blood pressure).
Lines 250-254: Why do you mention renal veins here? Do they participate in ultrafiltration? Under the subtitle “Glomerular Cxs and Panxs-based channels”, the reader expect to find references related to structures included in the renal glomeruli and involved with the process of ultrafiltration and not from other sections of the nephron or from the general vasculature of the kidney.
Line 273:…” No data of Panx2 this expression”: The term “this” is OK here?
Line 274: ..”Panx1 channel opening induces” …: include the site (cellular type) where these Panx1 channels are present, i.e., in renin-expressing cells (?)
Lines 277-278:….”..show expression in the renal vasculature where the P2X1 receptor increases renal vascular resistance…” and ..” the P2X1 receptor increases renal vascular resistance “: The expressions "renal vasculature" and “renal vascular resistance” are too general, are you always referring to the afferent arteriole? Indeed, changes in glomeruli arteriole have not the same consequences in ultrafiltration as compared to the efferent arteriole or the capillaries. In addition, are the effects of P1, P2X and P2Y related to Panx1 functional expression?
Tubular connexin and pannexin-based channels
Line 297: Replace "regulating" natriuresis by "increasing" or similar
Lines 303 - 305: Describing the expression and eventual roles of VASCULAR Panx1 under the subtitle "TUBULAR" Cxs and Panx1-channels is confusing. In line with this, juxtaglomerular cells consist on specialized smooth muscle cells of afferent arterioles; therefore, one expects this reference under the previous subtitle: "Glomerular".
Lines 309 -311: How would Panx1 regulate water permeability at the proximal tube via AQPs?
Line 312: Explicit which tubule from the nephron contains principal and intercalated cells.
Figure 4 is extremely helpful. Define YGC in the figure legend. Do the afferent and efferent arterioles express the same Cxs and Panxs? In addition, it would be interesting to detail the expression of these proteins in vasa recta if there is information available. The renal vasculature is a specialized one to permit the nephron accomplish its functional processes (ultrafiltration, reabsortion, etc) to generate the final product, the urine. Vessels from the glomerulus have different structure/function than the ones that run along tubules or the peritubular capillaries; do they express same Cxs and Panxs?
4. Cx43 and Panx1 in hypertensive nephropathy
It would be helpful to generate a table including different categories: the model employed to induce renal hypertension, the alteration (increase or decrease) of Cx43, the renal site expressing the altered Cx43, etc, references
Line 332: define "Aldo"
Line 335: where is the augmented Cx43 expressed exactly?
Line 346: where is the increased Cx43 expressed exactly?
Line 354: delete retinopathy
Line 393: Is here 162 the correct reference?
Lines 395-403: Why not to consider in the text the possibility that ATP is released via Cx43HCs as represented in the scheme (10.1523/JNEUROSCI.5048-07.2008)?
Line 406: Where was the concentration of AngII measured?
Line 407: Where are AT1 receptors expressed? Explicit
Lines 415-417: Can you speculate how alteration in mesangial cells function would affect glomerular ultrafiltration? See The Kidney from Brenner & Rector´s. file:///D:/Usuario/Desktop/TEXTOS%20PDF/Brenner%20&%20Rector's%20-%20The%20Kidney.pdf
5. Conclusions and Perspectives
Lines 427 - 428: Which kidney functions? You begin the sentence as "In particular" but the at the end the proposal is too general
Lines 439 - 440: Instead of HCs and pannexons, replace by one of these possibilities: connexons/pannexons, Cxs/Panxs HCs or CxHCs and pannexons

Author Response
REVIEWER 1
In this review, the authors describe the expression of Cxs and Panx1-based hemichannels (HCs) (also known as connexons and pannexons) by the kidney and revise the involvement of these HCs in renal damage during hypertensive nephropathy. They elaborate a hypothesis focusing on the deleterious effects of HCs expressed by mesangial cells during renal hypertension where the roles of angiotensin II, ATP and calcium ions are highlighted. The review is timely and interesting and raises a topic that is increasingly addressed by the scientific community. Likewise, although described injurious processes are framed in the context of hypertensive nephropathy, they could represent a common mechanism underlying several nephropathies. Anyway, although the manuscript contains valuable information, there are some conceptual aspects described in the text (especially in the introduction) that should be improved to generate an acceptable manuscript since they reflect whether the authors understand key aspects of renal physiology.
We appreciate the detailed comments and corrections, and we have made changes accordingly.
Abstract
The abstract contains many details (which are subsequently developed in the text) that distract from main messages; in fact, it would be preferable to summarize the background and to include the aim of the review at the end of the abstract.
R: Thank you very much for your comment. In this new version of our manuscript, we improved the abstract according to the reviewer’s comment.
- Introduction
The content described in lines 42-70 should be re-written or almost deleted. This is the main weakness of the manuscript. The kidney is a vital organ that regulates homeostasis of the internal media. Be aware that, “ultrafiltration, reabsorption, secretion and excretion” represent the functional processes developed by the nephron (the functional unit of the kidney) to maintain the homeostasis of the internal media. In line with this, the kidney eliminates those products that are required to be removed from the body and conserves the ones necessary to keep in order to successfully accomplishing the maintenance of internal medium homeostasis (volume, osmolality, arterial pressure, etc). Functional processes occurring in the nephron (ultrafiltration, reabsorption, secretion and excretion) and homeostatic functions of the kidney (regulation of the internal milieu including pH, osmolality, volume, arterial pressure) are not rigorously differentiated over the text. In addition, the description of renal structures should follow a logical sequence; i.e., lines 48-49: “Each nephron consists of a glomerulus, proximal and distal convoluted tubule, and loop of Henle.” Why to mention loop of Henle after distal convoluted tubule? Is the information about cortical and juxtaglomerular nephrons necessary for the revision (lines 51-53)? Similar question about the description about the renal epithelium (lines 62-68). These descriptions focus on some structural aspects, not related to the function they achieve and are incompletely assessed, indeed, it would be very difficult to resume and hierarchize the kidney structure and function in few paragraphs.
One possibility is to initiate the review by line 69, describing main homeostatic functions of the kidney (“regulation of the homeostasis of the internal media”, i.e., regulation of the volume, hidro-electric composition, osmolality, pH, arterial pressure) and then introducing to CKD, hypertension, role of ANGII and finally describe mesangial cells. Accordingly, the content of lines 54-61 should be relocated when referring to MCs.
Consider revising and citing: 10.1161/CIRCRESAHA.121.318064
R: Thank you very much for your comment. In this new version of our manuscript, we restructured and reshaped the introductory text to increase its clarity and scope.
- Connexins and Pannexins
- Connexins and Pannexins-based channels in the kidney
Glomerular connexin and pannexin-based channels
Lines 245-248: functional processes of the nephron (reabsorption and secretion of metabolites, glomerular filtration, water reabsorption, and renin secretion) should be differentiated from general homeostatic functions of the kidney (acid-base homeostasis; regulation of blood pressure).
R: Thank you for your comment. In this new version of our manuscript, we made that difference between the functions of the nephron and its homeostatic functions.
Lines 250-254: Why do you mention renal veins here? Do they participate in ultrafiltration? Under the subtitle “Glomerular Cxs and Panxs-based channels”, the reader expect to find references related to structures included in the renal glomeruli and involved with the process of ultrafiltration and not from other sections of the nephron or from the general vasculature of the kidney.
R: Thank you very much for your comment. In this new version of our manuscript, we focuse on showing the participation of connexins and pannexins in the glomerulus in general. What you mention about the renal vasculature, this is part of the beginning of this title and is part of figure 4.
Line 273:…” No data of Panx2 this expression”: The term “this” is OK here?
R: Thank you very much for your comment. In this new version of our manuscript, we modified the text according to the reviewer’s suggestion
Line 274: ..”Panx1 channel opening induces” …: include the site (cellular type) where these Panx1 channels are present, i.e., in renin-expressing cells (?)
R: Thank you very much for your comment. In this new version of our manuscript, we modified the text according to the reviewer’s suggestion.
Lines 277-278:….”..show expression in the renal vasculature where the P2X1 receptor increases renal vascular resistance…” and ..” the P2X1 receptor increases renal vascular resistance “: The expressions "renal vasculature" and “renal vascular resistance” are too general, are you always referring to the afferent arteriole? Indeed, changes in glomeruli arteriole have not the same consequences in ultrafiltration as compared to the efferent arteriole or the capillaries. In addition, are the effects of P1, P2X and P2Y related to Panx1 functional expression?
R: Thank you very much for your comment and in this new version what you mention has already been corrected. We hope that in these lines we answered your questions.
Regarding your query the effects of P1, P2X and P2Y related to Panx1 functional expression; it has been observed as an alternative for the release of nucleotides from cells through a non-specific mechanism (doi: 10.1016/j.imlet.2018.11.006).
Tubular connexin and pannexin-based channels
Line 297: Replace "regulating" natriuresis by "increasing" or similar
R: Thank you very much for your comment. In this new version of our manuscript, we modified the text according to the reviewer’s suggestion
Lines 303 - 305: Describing the expression and eventual roles of VASCULAR Panx1 under the subtitle "TUBULAR" Cxs and Panx1-channels is confusing. In line with this, juxtaglomerular cells consist on specialized smooth muscle cells of afferent arterioles; therefore, one expects this reference under the previous subtitle: "Glomerular".
R: Thank you very much for your comment. In this new version of our manuscript, we modified the text according to the reviewer’s suggestion.
Lines 309 -311: How would Panx1 regulate water permeability at the proximal tube via AQPs?
R: Regarding your question How would Panx1 regulate water permeability at the proximal tube via AQPs. When reviewing this information to date, nothing has been observed in this regard in the proximal tubule. What has been seen is its action on the function of AQ2 in cortical collecting ducts (CCD) cells, where it has been seen that AQP2 by modulating Ca2+, ATP and Panx1 differently could explain AQP2-increased cell migration in these cells (doi: 10.1002/jcp.30013). For this reason, in this new version what you mention has already been corrected.
Line 312: Explicit which tubule from the nephron contains principal and intercalated cells.
R: Thank you very much for your comment. In this new version of our manuscript, we modified the text according to the reviewer’s suggestion
Figure 4 is extremely helpful. Define YGC in the figure legend. Do the afferent and efferent arterioles express the same Cxs and Panxs? In addition, it would be interesting to detail the expression of these proteins in vasa recta if there is information available. The renal vasculature is a specialized one to permit the nephron accomplish its functional processes (ultrafiltration, reabsortion, etc) to generate the final product, the urine. Vessels from the glomerulus have different structure/function than the ones that run along tubules or the peritubular capillaries; do they express same Cxs and Panxs?
R: Thank you very much for your comment. In this new version of our manuscript, we included a detailed information regarding the specific expression of connexins and pannexins in afferent and efferent arterioles, and vasa recta. You can find these changes between lines 314 and 318.
Regarding your query, only the expression of Cx37 and Cx43 have been observed in the peritubular capillaries (doi: 10.1152/ajprenal.00255.2010).
- Cx43 and Panx1 in hypertensive nephropathy
It would be helpful to generate a table including different categories: the model employed to induce renal hypertension, the alteration (increase or decrease) of Cx43, the renal site expressing the altered Cx43, etc, references.
R: Thank you very much for your comment. In this new version of our manuscript, we included the suggested table detailing changes in expression for Cx43 in several renal areas
Line 332: define "Aldo"
R: Thank you very much for your comment. In this new version of our manuscript, we amended the text according to the reviewer’s concern
Line 335: where is the augmented Cx43 expressed exactly?
R: Thank you very much for your comment. We included the information according to the reviewer’s comment
Line 346: where is the increased Cx43 expressed exactly?
R: Thank you very much for your comment. We included the information according to the reviewer’s comment
Line 354: delete retinopathy
R: The text was modified according to the reviewer’s comment
Line 393: Is here 162 the correct reference?
R: The text was modified according to the reviewer’s concern
Lines 395-403: Why not to consider in the text the possibility that ATP is released via Cx43HCs as represented in the scheme (10.1523/JNEUROSCI.5048-07.2008)?
R: Thank you very much for your comment. In this new version of our manuscript, we included the possibility raised by the reviewer along with the reference suggested.
Line 406: Where was the concentration of AngII measured?
R: Thank you very much for your comment. We included the information according to the reviewer’s comment
Line 407: Where are AT1 receptors expressed? Explicit
R: Thank you very much for your comment. We included the information according to the reviewer’s comment
Lines 415-417: Can you speculate how alteration in mesangial cells function would affect glomerular ultrafiltration? See The Kidney from Brenner & Rector´s. file:///D:/Usuario/Desktop/TEXTOS%20PDF/Brenner%20&%20Rector's%20-%20The%20Kidney.pdf
R: Thank you very much for your observation. The idea presented in the text was revisited and rewritten considering the possible role of mesangial cells in glomerular ultrafiltration and how this could impact the development of kidney damage.
- Conclusions and Perspectives
Lines 427 - 428: Which kidney functions? You begin the sentence as "In particular" but the at the end the proposal is too general
R: Thank you very much for your comment. We amended the text according to the reviewer’s comment
Lines 439 - 440: Instead of HCs and pannexons, replace by one of these possibilities: connexons/pannexons, Cxs/Panxs HCs or CxHCs and pannexons
R: Thank you very much for your comment. We amended the text according to the reviewer’s comment

Reviewer 2 Report
In this manuscript, Lucero, et al. summerize the roles and mechanisms of Cx hemichannels and pannexons in hypertensive nephrology. In general, the authors address a currently exciting topic. The manuscript, in large part, is well-written. The schematic depiction is well-made and helpful in understanding the content. However, the manuscript also suffers from several weaknesses, as below.
Major comments
1) The basic renal structure and function description is too long and general. Given that mesangial cells and their regulation by HCs and pannexin are the review's focus, the authors should provide a more detailed description of the pathophysiological roles of mesangial cells. Furthermore, the pathophysiological roles of juxtaglomerular appratus should also be mentioned because of its pivotal role in the renin-angiotensin-aldosterone system and the development of hypertension. With these backgrounds, the authors could easily understand the roles of HCs and pannexin in hypertensive nephropathy.
2) The manuscript needs improvement in focus and organization. For example, what are the main targets (mesangial cells, juxtaglomerular appratus?) and the mechanisms (Ang II, calcium, ATP, ROS, inflammation and cell phenotypes) of channels and how can they be linked to hypertensive nephropathy? These questions should be described more clearly.
3) Some description of kidney diseases needs to be checked for correctness, such as lines 83-86 and Fig. 2 (reduced permeability, and filtration (tubular and glomerular). Usually, tubular cells are not involved in filtration and permeability.
4) Inappropriate expressions and typos need to be corrected (such as line 378 moderated increase; line 146, the long-term onset of CKD, etc.)In this manuscript, Lucero, et al. summerize the roles and mechanisms of Cx hemichannels and pannexons in hypertensive nephrology. In general, the authors address a currently exciting topic. The manuscript, in large part, is well-written. The schematic depiction is well-made and helpful in understanding the content. However, the manuscript also suffers from several weaknesses, as below.
Major comments
1) The basic renal structure and function description is too long and general. Given that mesangial cells and their regulation by HCs and pannexin are the review's focus, the authors should provide a more detailed description of the pathophysiological roles of mesangial cells. Furthermore, the pathophysiological roles of juxtaglomerular appratus should also be mentioned because of its pivotal role in the renin-angiotensin-aldosterone system and the development of hypertension. With these backgrounds, the authors could easily understand the roles of HCs and pannexin in hypertensive nephropathy.
2) The manuscript needs improvement in focus and organization. For example, what are the main targets (mesangial cells, juxtaglomerular appratus?) and the mechanisms (Ang II, calcium, ATP, ROS, inflammation and cell phenotypes) of channels and how can they be linked to hypertensive nephropathy? These questions should be described more clearly.
3) Some description of kidney diseases needs to be checked for correctness, such as lines 83-86 and Fig. 2 (reduced permeability, and filtration (tubular and glomerular). Usually, tubular cells are not involved in filtration and permeability.
4) Inappropriate expressions and typos need to be corrected (such as line 378 moderated increase; line 146, the long-term onset of CKD, etc.)
Author Response
REVIEWER 2
Major comments
1) The basic renal structure and function description is too long and general. Given that mesangial cells and their regulation by HCs and pannexin are the review's focus, the authors should provide a more detailed description of the pathophysiological roles of mesangial cells. Furthermore, the pathophysiological roles of juxtaglomerular apparatus should also be mentioned because of its pivotal role in the renin-angiotensin-aldosterone system and the development of hypertension. With these backgrounds, the authors could easily understand the roles of HCs and pannexin in hypertensive nephropathy.
R: Thank you very much for your comment. In this new version of our manuscript, we restructured and reshaped the introductory text to increase its clarity and scope: mesangial cells and juxtaglomerular apparatus
2) The manuscript needs improvement in focus and organization. For example, what are the main targets (mesangial cells, juxtaglomerular apparatus?) and the mechanisms (Ang II, calcium, ATP, ROS, inflammation, and cell phenotypes) of channels and how can they be linked to hypertensive nephropathy? These questions should be described more clearly.
R: Thank you very much for your comment. In this new version of the manuscript, we rearranged the text to improve the focus and organization of the significant ideas. We detailed the primary targets and molecular mechanisms possibly involved in hypertensive nephropathy, particularly on hemichannels and pannexons.
3) Some description of kidney diseases needs to be checked for correctness, such as lines 83-86 and Fig. 2 (reduced permeability, and filtration (tubular and glomerular). Usually, tubular cells are not involved in filtration and permeability.
R: Thank you very much for your comment. We amended the text according to the reviewer’s concern
4) Inappropriate expressions and typos need to be corrected (such as line 378 moderated increase: line 146, the long-term onset of CKD, etc.)
R: Thank you very much for your comment. We amended the text according to the reviewer’s concern

Round 2
Reviewer 1 Report
This second version of the manuscript has been substantially improved.
Lines 38-39:
Authors: Kidney-mediated blood filtering maintains pH, water and electrolyte balance, thus contributing to regulating blood pressure.
R: Kidney-mediated blood filtering and modification of the ultrafiltration content by reabsorption and secretion, maintains pH, water and electrolyte balance, thus contributing to regulating blood pressure and osmolality.
Lines 76 - 79:
Authors: Based on the renin localization, mainly in granular cells of the renal afferent arteriole, it was assumed that swelling or shrinking of the juxtaglomerular apparatus (JGA) is involved in the glomerular blood flow ascribed to the renin-angiotensin system (RAS) based on the renin localization in granular cells of renal afferent arterioles.
R: Sentences in black are repeated.
Lines 246 - 250:
Authors: The evidence indicates that these connexins could play a pivotal role in important aspects of renal function, such as the maintenance of acid-base homeostasis, regulation of blood pressure, and functional processes of the nephron: reabsorption and secretion of metabolites, glomerular filtration, water reabsorption, electrolyte balance, and renin secretion.
R: The evidence indicates that these connexins could play a pivotal role in important aspects of renal function, such as the maintenance of acid-base homeostasis, hydroelectrolyte balance, regulation of blood pressure, and functional processes of the nephron: glomerular filtration, reabsorption and secretion of metabolites, water reabsorption and renin secretion.
Lines 309-310:
Authors: Panx1 also is expressed in the principal and intercalated cells in the distal tubule, which regulate water reabsorption and acid-base balance, respectively.
R: Panx1 also is expressed in the principal and intercalated cells in the collecting tubule, which regulates water reabsorption and acid-base balance, respectively.
Author Response
REVIEWER 1
This second version of the manuscript has been substantially improved.
We appreciate the detailed comments and corrections, and we have made changes accordingly.
Lines 38-39:
Authors: Kidney-mediated blood filtering maintains pH, water, and electrolyte balance, thus contributing to regulating blood pressure.
R: Kidney-mediated blood filtering and modification of the ultrafiltration content by reabsorption and secretion, maintains pH, water, and electrolyte balance, thus contributing to regulating blood pressure and osmolality.
R: Thank you very much for your comment. In this new version of our manuscript, we modified the text according to the reviewer’s suggestion
Lines 76 - 79:
Authors: Based on the renin localization, mainly in granular cells of the renal afferent arteriole, it was assumed that swelling or shrinking of the juxtaglomerular apparatus (JGA) is involved in the glomerular blood flow ascribed to the renin-angiotensin system (RAS) based on the renin localization in granular cells of renal afferent arterioles.
R: Sentences in black are repeated.
R: Thank you very much for your comment. In this new version of our manuscript, we modified the text according to the reviewer’s suggestion
Lines 246 - 250:
Authors: The evidence indicates that these connexins could play a pivotal role in important aspects of renal function, such as the maintenance of acid-base homeostasis, regulation of blood pressure, and functional processes of the nephron: reabsorption and secretion of metabolites, glomerular filtration, water reabsorption, electrolyte balance, and renin secretion.
R: The evidence indicates that these connexins could play a pivotal role in important aspects of renal function, such as the maintenance of acid-base homeostasis, hydroelectrolyte balance, regulation of blood pressure, and functional processes of the nephron: glomerular filtration, reabsorption and secretion of metabolites, water reabsorption and renin secretion.
R: Thank you very much for your comment. In this new version of our manuscript, we modified the text according to the reviewer’s suggestion
Lines 309-310:
Authors: Panx1 also is expressed in the principal and intercalated cells in the distal tubule, which regulate water reabsorption and acid-base balance, respectively.
R: Panx1 also is expressed in the principal and intercalated cells in the collecting tubule, which regulates water reabsorption and acid-base balance, respectively.
R: Thank you very much for your comment. In this new version of our manuscript, we modified the text according to the reviewer’s suggestion

Reviewer 2 Report
In this version of manuscript, my concerns have been addressed by the authors. However, there are still many inaccurate descriptions that needs to be corrected. The manuscript still needs improvement in concise and clarity.
For examples:
Lines 146 ~ 148, "thus triggering a reduction in tubular and glomerular permeability and filtration. Consequently, glomerular sclerosis and hypertensive nephropathy can contribute to both the perpetuation of renal damage and CKD" >> tubular cells are not involved in permeability and filtration. In addition, the next sentense is also difficult to understand the relationship and meaning.
Line 104, thus, ... >> the conjunction word thus is not used appropriately here.
There are many similar expressions, please have a check and correct them.
Author Response
REVIEWER 2
In this version of manuscript, my concerns have been addressed by the authors. However, there are still many inaccurate descriptions that needs to be corrected. The manuscript still needs improvement in concise and clarity.
We appreciate the detailed comments and corrections, and we have made changes accordingly.
For examples:
Lines 146 ~ 148, "thus triggering a reduction in tubular and glomerular permeability and filtration. Consequently, glomerular sclerosis and hypertensive nephropathy can contribute to both the perpetuation of renal damage and CKD" >> tubular cells are not involved in permeability and filtration. In addition, the next sentense is also difficult to understand the relationship and meaning.
R: Thank you very much for your comment. In this new version of our manuscript, we modified the text according to the reviewer’s suggestion
Line 104, thus, ... >> the conjunction word thus is not used appropriately here.
There are many similar expressions, please have a check and correct them.
R: Thank you very much for your comment. We amended the text according to the reviewer’s concern
